# Colorimetric RT-LAMP Detection of Multiple SARS-CoV-2 Variants and Lineages of Concern Direct from Nasopharyngeal Swab Samples without RNA Isolation

**DOI:** 10.3390/v15091910

**Published:** 2023-09-12

**Authors:** Santiago Werbajh, Luciana Larocca, Carolina Carrillo, Fabiana Stolowicz, Lorena Ogas, Sergio Pallotto, Solange Cassará, Liliana Mammana, Inés Zapiola, María Belén Bouzas, Adrian A. Vojnov

**Affiliations:** 1Instituto de Ciencia y Tecnología Dr. César Milstein, Fundación Pablo Cassará, CONICET. Saladillo 2468, Buenos Aires C1440FFX, Argentinaccarrillo@fundacioncassara.org.ar (C.C.); scassara@lpc.com.ar (S.C.); 2Laboratorio Pablo Cassará S.R.L. Saladillo 2452, Buenos Aires C1440FFX, Argentina; 3Sección Virología, Hospital de Enfermedades Infecciosas Francisco Javier Muñiz Uspallata 2272, Buenos Aires C1282AEN, Argentinazapiola.ines@gmail.com (I.Z.); mariabbouzas@gmail.com (M.B.B.); 4Facultad de Medicina-Universidad del Salvador, Av. Córdoba 1601, Buenos Aires C1055AAG, Argentina

**Keywords:** COVID-19, SARS-CoV-2, LAMP

## Abstract

Since, during the Coronavirus disease 19 (COVID-19) pandemic, a large part of the human population has become infected, a rapid and simple diagnostic method has been necessary to detect its causative agent, the Severe Acute Respiratory Syndrome-related Coronavirus-2 (SARS-CoV-2), and control its spread. Thus, in the present study, we developed a colorimetric reverse transcription-loop-mediated isothermal amplification (RT-LAMP) kit that allows the detection of SARS-CoV-2 from nasopharyngeal swab samples without the need for RNA extraction. The kit utilizes three sets of LAMP primers targeting two regions of ORF1ab and one region in the E gene. The results are based on the colorimetric change of hydroxynaphthol blue, which allows visual interpretation without needing an expensive instrument. The kit demonstrated sensitivity to detect between 50 and 100 copies of the viral genome per reaction. The kit was authorized by the National Administration of Drugs, Food and Technology (ANMAT) of Argentina after validation using samples previously analyzed by the gold standard RT-qPCR. The results showed a sensitivity of 90.6% and specificity of 100%, consistent with conventional RT-qPCR. In silico analysis confirmed the recognition of SARS-CoV-2 variants of concern (B.1.1.7, B.1.351, P.1, B.1.617.2, B.1.427, and B.1.429), and lineages of the Omicron variant (B.1.1.529) with 100% homology. This rapid, simple, and sensitive RT-LAMP method paves the way for a large screening strategy to be carried out at locations lacking sophisticated instrumental and trained staff, as it particularly happens in regional hospitals and medical centers from rural areas.

## 1. Introduction

Coronavirus disease 19 (COVID-19) is an acute, sometimes severe, respiratory illness caused by the Severe Acute Respiratory Syndrome-related Coronavirus—2 (SARS-CoV-2). The presence of COVID-19 was first reported in Wuhan, China, in late 2019, and since then, the infection has spread widely around the world. The most common method for nucleic acid diagnosis of COVID-19 is based on real-time PCR (RT-PCR) [1,2,3]. Although RT-PCR is sensitive and reliable, it is time-consuming (~2 to 4 h) and requires a specific detection device or instrument, which limits its broad application to the huge demand due to the global pandemic of COVID-19.

Loop-mediated isothermal amplification (LAMP) is a rapid technology of DNA amplification [4,5,6], which has been applied for the detection of different pathogens, such as viruses, bacteria, and parasites [4,7,8,9,10]. The LAMP reaction is often performed at a constant temperature, and the target DNA can be amplified in 30 min [5]. Initially, the LAMP reactions used four primers, and later, results showed that the use of two additional loop primers allowed shortening by half the time required for the original LAMP reaction [11]. Thus, the LAMP method uses 4–6 primers to bind 6–8 regions of a target DNA, and its specificity is extremely high [5,6]. Since SARS-CoV-2 has an RNA length of approximately 30 kb [12,13,14], a single step of reverse transcription RT-LAMP reactions can be carried out together, significantly shortening the time needed. Reverse transcription and LAMP can be combined using Warm Start RTx Reverse Transcriptase (New England Biolabs, Ipswich, UK).

Various methods have been developed for the visual detection of LAMP products [15]. Among them, The SYBR Green intercalating dye method has been used for SARS-CoV-2 detection [16]. However, it is important to note that the use of this dye may introduce interference in the LAMP reaction and the need to open reaction tubes for the addition of DNA-binding dyes after LAMP completion, which increases the risk of contamination and, therefore, potentially leads to false positives [9,17,18]. On the other hand, turbidity-based methods exploit the formation of magnesium pyrophosphate (Mg_2_P_2_O_7_) precipitate, a byproduct of the LAMP reaction, resulting in enhanced turbidity [7,19]. During the formation of Mg_2_P_2_O_7_ within the LAMP reaction, the Mg^2+^ ions undergo reduction. Consequently, the use of a metal ion indicator dye to monitor Mg^2+^ ions emerges as an alternative approach to detect positive LAMP reactions and has thus been applied for COVID-19 detection. These metal ion indicator dyes exhibit color changes in the presence or absence of divalent metal ions, a phenomenon discernible to the naked eye or with basic instrumentation [5,20]. One of these dyes is hydroxynaphthol blue (HNB), which shifts from violet to sky blue as the LAMP reaction progresses [20]. This dye has been used to detect SARS-CoV-2 viral RNA in blood samples extracted by the TRIzol method [21]. Positive LAMP reactions can also be visually identified using calcein dye, another metal ion indicator, which shifts color from orange to fluorescent green due to a reduction in the Mg^2+^ ions concentration, particularly used in SARS-CoV-2 detection [5,22]. Both HNB and calcein dyes can be introduced in a pre-reaction step, obviating the need to open the tube after the LAMP reaction, thereby averting subsequent contamination.

Multiple SARS-CoV-2 variants have been described. For example, the variant known as 20I/501Y.V1, VOC 202012/01, or B.1.1.7 [23], with a large number of mutations, emerged in the UK and was then detected in several countries around the world. Scientists from the UK reported evidence suggesting that this variant may be associated with an increased risk of death compared with other variants [24]. In South Africa, another variant of SARS-CoV-2 (known as 20H/501Y.V2 or B.1.351) emerged independently of B.1.1.7 but shares some mutations with it. Cases attributed to this variant have been detected in multiple countries outside South Africa [25]. On the other hand, the variant known as P.1, which has 17 unique mutations, including three in the receptor binding domain of the spike protein [26], was first identified in four travelers from Brazil who were tested during routine screening at Haneda airport outside Tokyo, Japan. The California or West Coast variants (B.1.427 and B.1.429 lineages) likely emerged in late spring or early summer of 2020. These two variants carry similar, although slightly different, genetic mutations [26]. Finally, the emergence of the Omicron variant has raised significant concerns worldwide. This new variant, first identified in November 2021, has demonstrated an unprecedented number of mutations in its spike protein, which plays a crucial role in viral entry and infectivity. The Omicron variant exhibits more than 50 mutations in the spike protein, including several deletions and insertions, resulting in potential changes in its antigenic properties [27] and an increase in transmissibility compared to previous variants [28]. Moreover, studies indicate a potential reduction in the neutralizing antibodies generated by previous infection or vaccination against the Omicron variant [29].

In the present study, we developed a simple COVID-19 diagnosis kit for the rapid detection of SARS-CoV-2, using one-step reverse transcription and loop-mediated isothermal amplification (RT-LAMP) directly from nasopharyngeal swabs. The entire process takes only 60 min at a constant temperature of 64 °C, with no need for complex equipment. The detection limit is 100 copies of viral RNA per reaction. The simple color change can be visualized by the naked eye to confirm the result of specific viral RNA sequence detection. Our kit was validated by clinical COVID-19 samples, and in silico analysis confirmed the detection of all circulating variants, including the Omicron variant.

## 2. Methods

### 2.1. Design of RT-LAMP Primers for SARS-CoV-2 Genome Detection

Three sets of LAMP primers based on the Wuhan-Hu-1 SARS-CoV-2 complete genome sequence (GenBank: MN908947.3) [30]. were designed to target the open reading frame ORF1aa, open reading frame ORF1ab and envelope (E) gene, respectively (Table 1). The primers were designed using the Primer Explorer Version 5 (Eiken Chemical Co., Ltd., Tokyo, Japan). The loop primers (LF and LB) were manually designed, ensuring appropriate melting temperatures, using the Vector NTi Advance 10.1.1 windows software (Invitrogen, Life technologies Corporation. Carlsbad, CA, USA).

The amplicon sizes of ORF1aa, ORF1ab, and E were 220 bp, 289 bp, and 215 bp, respectively. All the primers were validated using Quantitative Synthetic SARS-CoV-2 RNA (ATCC® VR-3276SD™), and oligos were ordered from Macrogen Inc., Seoul, Korea, and resuspended in UltraPure water at a concentration of 100 μM. The localization of the LAMP target regions on the SARS-CoV-2 genome is illustrated in Figure 1 and Appendix A.

### 2.2. Sample Inactivation and Treatment for Direct Assay

Nasopharyngeal swab samples in viral transport medium (0.9% NaCl) were obtained from the Virology Unit of the Infectious Diseases Hospital “Francisco Javier Muñiz”, Buenos Aires, Argentina, and analyzed by RT-PCR (GeneFinder) after automatized RNA purification (Perkin Elmer, Waltham, MA, USA). The same samples were assessed by direct RT-LAMP without RNA extraction. For RT-LAMP, 10 µL of the swab samples were pre-treated at 95 °C for 8 min with 40 µL of lysis buffer A (Tris-HCl pH 8.8, 20 mM; EDTA, 0.1 mM) and lysis buffer B (Tris(2-carboxyethyl) phosphine hydrochloride 2.5 mM) at a 180:1 (*v*/*v*) ratio. The samples were then briefly centrifuged, and 10 µL was used for the subsequent reaction.

### 2.3. RT-LAMP Assay

The RT-LAMP assay was performed using a dry thermal block with a 0.5 mL PCR tube holder. The final LAMP conditions included 1.6 µM of each FIP and BIP primer, 0.2 µM of each F3 and B3 primer, 0.4 µM of each LF and LB loop primers. 8 U of Bst DNA polymerase large fragment (New England Biolabs. Ipswich, UK), 7.5 U of WarmStartRTx Reverse Transcriptase (New England Biolabs), 120 µM HNB (Sigma-Aldrich, St. Louis, MO, USA), 8 mM MgSO_4_, 1.4 mM of dNTP mix, 20 mM Tris-HCl (pH 8.8), 10 mM KCl, 10 mM (NH_4_)_2_SO_4_, 0.1% tween-20 and 1 M betaine, in a final volume of 40 μL, including the template (30 µL reaction buffer and 10 µL of sample template). The reaction mixture was incubated at 64 °C for 60 min. The amplification products were visualized by the color change from violet to blue-sky blue, indicating the presence of the target viral RNA sequence. In some cases, the products were also analyzed by electrophoresis on a 2% agarose gel.

### 2.4. Primer Cross-Reactivity (In Silico Sequence Analysis)

To evaluate the analytical specificity of the RT-LAMP and cross-reactivity among primers, BLASTN analysis of the three sets of six primers was performed against a high-priority pathogen database that included other closely related coronaviruses available in the public domain NCBI Nucleotide database under the context of GenBank Release 240.0 (consisting of 219,055,207 sequences with a total of 698,688,094,046 bases, including 1,947,019,989 sequences with 9,627,627,030,647 bases for traditional GenBank records and sequences from set-based records). The search was performed with the BLASTN 2.9.0+ software. Latest inclusivity data were collected from whole genome sequences of SARS-CoV-2 published via GISAID (www.gisaid.org. Accessed on 3 March 2023) and whole genome sequences published via the National Center for Biotechnology Information (www.ncbi.nlm.nih.gov-2020) (Appendix A).

### 2.5. Analytical Specificity of the RT-LAMP Assay

Cross-specificity tests for the multiplex RT-LAMP assay were carried out using purified nucleic acid (Qiagen) of viral and bacterial pathogens associated with respiratory infections, comprising influenza A, influenza B, Canine coronavirus, Dengue 1-4, Chikungunya, Yellow Fever, Zika, Mayaro, Mpox, Hela RNA, *Trypanosoma cruzi*, *Mycobacterium tuberculosis*, *Escherichia coli*, *Saccharomyces cerevisiae*, *Saccharomyces pombe*, and *Pichia pastoris*.

### 2.6. Repeatability, Reproducibility, and Detection Limit of the RT-LAMP Assay

Purified SARS-CoV-2 RNA (AmplirunSARS-CoV-2 RNA control, 12,000–20,000 copies/μL once reconstituted) was used as template in RT-LAMP reactions, with different operators, eight replicates at each template, and a no-target control. The detection limit was determined by conducting serial dilutions of viral genome copies, including 1000, 500, 100, 50, 25, 12, 6, and 0 copies per reaction.

### 2.7. Clinical Evaluation of the RT-LAMP Assay for SARS-CoV-2 Detection

To assess the performance of the RT-LAMP assay using the final combined primer sets, we used 192 samples obtained from nasopharyngeal swabs collected from patients at the Virology Section of the Infectious Diseases Hospital “Francisco Javier Muñiz”, Buenos Aires, Argentina. For virus inactivation, the samples were processed with lysis buffer and incubated at 98 °C for 8 min. These samples included samples from individuals who tested positive (*n* = 139) and negative (*n* = 53) for SARS-CoV-2 RNA based on RT-qPCR, which served as the gold standard.

## 3. Results

### 3.1. Design and Specificity of the ORF1aa, ORF1ab, and E Gene Primers for SARS-CoV-2 Target Sequence Identification

To identify the most specific target region for the design of LAMP primers, we used VectorNti Advance software (Invitrogen) to identify conserved genomic areas of SARS-CoV-2. Three conserved genomic regions were selected: two sequences from ORF1a (ORF1aa and ORF1ab), which encode the replicase polyprotein [31], and one sequence from the E gene, which encodes the envelope protein [32] (Figure 1). For each selected region, six primers were designed, targeting regions of approximately 240–260 bp in length, suitable for rapid amplification (Figure 1 and Table 1). Alignments of the primers against the SARS-CoV-2 reference genome (MN908947) showed 100 % identity, indicating the absence of mismatch (Appendix A). Additionally, in silico specificity analysis was performed using genome sequences from other coronavirus and respiratory pathogens, demonstrating no more than 80% homology between the designed primers and those genomes (Appendix A). Thus, no cross-reactivity is expected in this SARS-CoV-2 detection method.

### 3.2. Evaluation of Primer Annealing with SARS-CoV-2 Variants and Mutations

To investigate the impact of mutations on primer annealing, in silico analysis of cross-reactivity was performed using our designed primers with the following variants and mutations: the UK variant known as 501Y.V1, VOC 202012/01, or B.1.1.7 lineage, the South African variant known as 501Y.V2 or B.1.351 lineage, the Brazilian variant known as 501Y.V3 or P.1 lineage, the California variants known as B.1.427/ B.1.429, and the most recent lineages of the Omicron variant (B.1.1.529, BA.1, BA.1.1, BA.2, BA.3, BA.4, and BA.5). The analysis, based on the latest inclusivity data from GISAID and NCBI, confirmed no mismatches between the primers and the target regions of the variants (Figure 1 and Appendix A).

### 3.3. Development of the RT-LAMP Assay and Evaluation of Its Analytical Sensitivity

The detection of SARS-CoV-2 RNA was assessed with the Quantitative SARS-CoV-2 RNA standard (Amplirun SARS-CoV-2 RNA control, VIRCELL). Figure 2 shows the specificity and the integration of reverse transcription and LAMP into a single RT-LAMP reaction to amplify viral RNA fragments specifically. This was achieved by using the set of three primers both separately and all together with the Quantitative SARS-CoV-2 RNA standard. Positive reactions resulted in a color change from violet to blue-sky-blue, indicating amplification of the target sequence by Bst DNA polymerase activity, while negative reactions retained the purple color (Figure 2A). The visible read-outs were consistent with the results obtained by gel electrophoresis (Figure 2B).

To determine the limit of detection (LOD) of the RT-LAMP assay, we next evaluated its sensitivity using the quantitative synthetic SARS-CoV-2 RNA standard. The copy number of RNA targets was diluted from 1000 to 6 copies per reaction to assess the assay sensitivity. The efficiency of the assay was determined by sampling the reaction mixture at 60 min using a pool of the three primer sets. Results showed that 25 copies of the target RNA were successfully amplified. However, a repeatability assay conducted by different operators demonstrated 100% amplification efficiency in samples with more than 50 copies (Figure 2C,D). This result indicated that the RT-LAMP assay tested was sensitive enough to detect viral RNA down to 50 copies per reaction.

### 3.4. Analytical Specificity

The analytical specificity of the assay was re-evaluated using a synthetic SARS-CoV-2 RNA standard, as well as RNA genomes from other viruses such as Influenza A (H1N1), Influenza B, Canine coronavirus, Dengue 1–4, Zika, Chikungunya, and Mayaro. Additionally, potential contaminants, including *Saccharomyces cerevisiae*, *Saccharomyces pombe*, *Escherichia coli*, and genomic human DNA, were included for testing using DNA from HeLa Cells. After incubating the samples for 60 min, the results indicated that the primers used in the kit specifically and accurately detected SARS-CoV-2 without any cross-reactivity (Figure 3).

### 3.5. Clinical Validation

The RT-LAMP assay developed to detect SARS-CoV-2 RNA was clinically validated from direct nasopharyngeal swab samples. The samples were pre-treated with lysis buffer and heat for 8 min at 98 °C. A test kit comprising tubes with pools of primers (ORF1aa, ORF1ab, and E primers) was used. Swab samples (139 positive and 53 negative samples) collected from patients following standard procedures were initially tested using conventional RT-qPCR. Subsequently, the same samples were subjected to RT-LAMP after the treatment described in the Methods section. The contingency table of the results, presented in Table 2, indicates that 126 out of the 139 samples identified as positive by RT-qPCR were also detected as positive by RT-LAMP. In other words, there were only 13 false negative results. In addition, the negative samples previously identified as negative using RT-qPCR were also found to be negative when tested with RT-LAMP.

The correlation observed between the cycle threshold (CT) values, which reflect the viral loads of the samples used in tests conducted through RT-PCR with purified RNA specimens, and the results achieved utilizing the RT-LAMP kit developed underscores the robust capability of the latter to consistently detect even relatively low viral concentrations within the same samples but without RNA purification. Nonetheless, instances of false negatives are apparent in samples exhibiting CT values around and above 30, indicating a viral load threshold in the samples below which the RT-LAMP kit may not be capable of detection (Appendix A).

The analysis of the results obtained comparing the RT-LAMP with RT-PCR as gold standard (Table 2) indicated that the RT-LAMP developed has a Positive Predictive Value (PPV) of 100%, which indicates that a positive result possesses a 100% probability that the patient is truly infected with SARS-CoV-2, and a Negative Predictive Value (NPV) of 80.3%, which indicates that when the RT-LAMP gives a negative result, there is an 80.3% probability that the patient is truly not infected with SARS-CoV-2.

Considering both PPV and NPV, the values obtained suggest a high confidence that a positive result with the RT-LAMP is reliable in correctly identifying infected patients and a high probability that a negative result is reliable in correctly ruling out the presence of the infection in a patient.

The results obtained demonstrated a clinical specificity of 100% and a clinical sensitivity of 90.6% (Table 2), indicating a strong agreement between the RT-LAMP assay and the conventional RT-qPCR.

## 4. Discussion

In this study, we developed and validated a colorimetric molecular isothermal detection assay for SARS-CoV-2 (RT-LAMP) from nasopharyngeal swabs, which includes a lysis buffer for sample treatment, enabling virus detection without the need for RNA purification. Results demonstrated its reliability as a diagnostic tool. Our findings reveal strong specificity, accurately identifying SARS-CoV-2 due to the strategic selection of three distinct conserved regions within the viral genome.

A significant advantage of our RT-LAMP kit lies in its operational simplicity and user-friendliness. The inclusion of HNB, a dye that does not interfere with the LAMP reaction, along with dNTPs, within a single reaction tube, combined with the arrangement of enzymes and other reagents within a dropper, allows a straightforward and rapid assay. By eliminating the need to open the reaction tube after the reaction, the risk of contamination is effectively minimized.

This unique design, coupled with the provision of a lysis buffer allowing for minimal sample treatment without viral RNA purification, makes this kit highly suitable for on-site healthcare settings and resource-limited environments.

The analytical sensitivity of determinations using RT-LAMP was demonstrated by its ability to detect viral RNA down to 100 copies per reaction. This sensitivity, combined with the capability to detect relatively low viral concentrations, enhances its potential for early infection detection and effective disease control. However, it is noteworthy that instances of false negatives were observed in samples with CT values around and above 30, indicating a potential limitation in detecting very low viral loads.

Compared with conventional RT-qPCR, the clinical validation of our RT-LAMP assay exhibited a clinical specificity of 100% and a clinical sensitivity of 90.67%. The PPV and NPV of the assay underscore its reliability in correctly identifying both infected and non-infected patients, further validating its utility as a diagnostic tool.

In summary, the combination of specificity, sensitivity, operational simplicity, and potential for point-of-care use establishes the RT-LAMP assay as a robust tool for SARS-CoV-2 detection. Its advantages render it particularly suitable for rapid and accurate testing, especially in settings where complex laboratory infrastructure is not easily accessible.

## Figures and Tables

**Figure 1 viruses-15-01910-f001:**
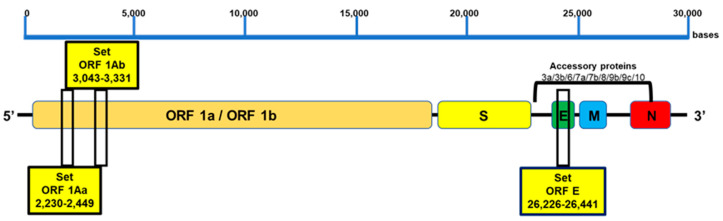
Target region for LAMP primer design. Three conserved genomic regions were selected: two sequences from ORF1a (ORF1aa and ORF1ab) and one sequence from the E gene with high homology among different SARS-CoV-2 variants. The target regions of approximately 240–260 bp in length are indicated.

**Figure 2 viruses-15-01910-f002:**
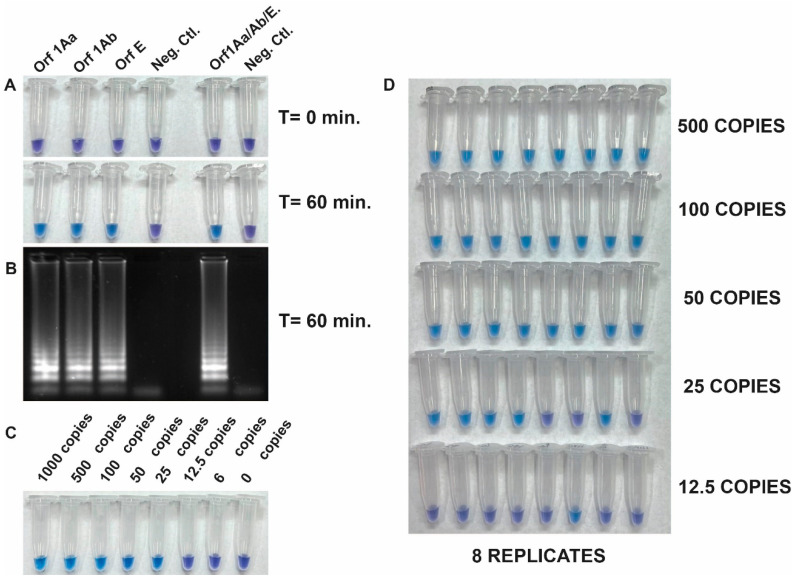
(**A**) Specificity and integration of reverse transcription and LAMP (RT-LAMP) in a single reaction. Specific amplification of a Quantitative SARS-CoV-2 RNA standard using a set of three primers individually and in combination. (**B**) Gel electrophoresis analysis of the same reaction tubes from panel A. (**C**) Determination of the limit of detection (LOD) of the RT-LAMP assay using dilution (from 1000 to 6 copies) of the quantitative synthetic SARS-CoV-2 RNA standard (AmplirunSARS-CoV-2 RNA control, VIRCELL) as RNA target and a pool of the three primer sets. (**D**) Repeatability assay conducted by different operators.

**Figure 3 viruses-15-01910-f003:**
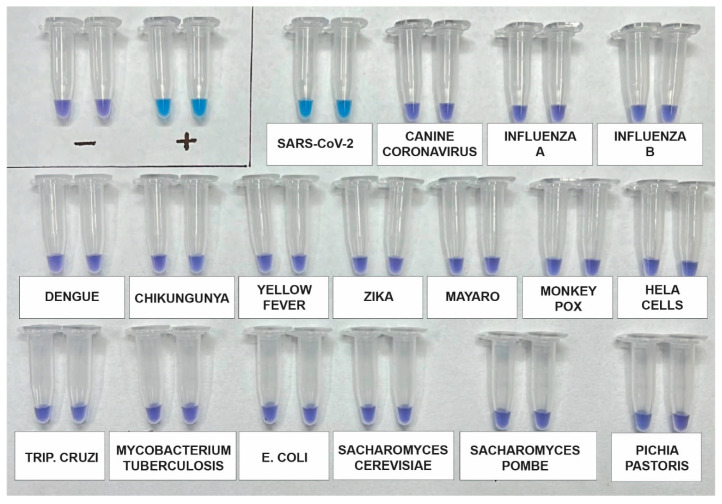
Analytical specificity assay using a synthetic SARS-CoV-2 RNA standard and RNA genomes from other viruses, including Influenza A (H1N1), Influenza B, Canine coronavirus, Dengue 1-4, Zika, Chikungunya, and Mayaro. Additionally, potential contaminants such as *Saccharomyces cerevisiae*, *Saccharomyces pombe*, *Escherichia coli*, and genomic human DNA from HeLa cells were included. The assay was performed in duplicate and across three independent tests.

**Table 1 viruses-15-01910-t001:** Design of LAMP primers for specific target regions in SARS-CoV-2. Six primers were designed for each of the three following regions: ORF1aa, ORF1ab, and E gene), targeting 240–260 bp using Vector NTi Advance software (Invitrogen).

Target	Primer	Sequence
ORF1aa	R1Aa-2F3	TGCTTGTGAAATTGTCGGT
	R1Aa-2B3	GCCAGTTTCTTCTCTGGAT
	R1Aa-2FIP	TCAGCACACAAAGCCAAAAATTTAT-AAATTGTCACCTGTGCAAAG
	R1Aa-2BIP	TATTGGTGGAGCTAAACTTAAAGCC-ACACTTTCTGTACAATCCCTT
	R1Aa-2LP	CTTAAAGAATGTCTGAACACTCTCC
	R1Aa-2LB	GAATTTAGGTGAAACATTTGTCACG
ORF1ab	R1Ab-1F3(R1AB)	TCCAGATGAGGATGAAGAAGA
	R1Ab-1B3	AGTCTGAACAACTGGTGTAAG
	R1Ab-1FIP	AGAGCAGCAGAAGTGGCACAGGTGATTGTGAAGAAGAAGAG
	R1Ab-1BIP	TCAACCTGAAGAAGAGCAAGAACTGATTGTCCTCACTGCC
	R1Ab-1LF	CTCATATTGAGTTGATGGCTCA
	R1Ab-1LB	ACAAACTGTTGGTCAACAAGAC
ORF E	pE-3F3(PE3)	AGCTGATGAGTACGAACTT
	pE-3B3	TTCAGATTTTTAACACGAGAGT
	pE-3FIP	ACCACGAAAGCAAGAAAAAGAAGTATTCGTTTCGGAAGAGACAG
	pE-3BIP	TTGCTAGTTACACTAGCCATCCTTAGGTTTTACAAGACTCACGT
	pE-3LF	ACGCTATTAACTATTAACGTAC
	pE-3LB	CTGCGCTTCGATTGTGTGCGT

**Table 2 viruses-15-01910-t002:** Clinical validation of the RT-LAMP assay developed for SARS-CoV-2 RNA detection on direct nasopharyngeal swab samples. The assay showed a specificity of 100% and a sensitivity of 90.6%.

	Reference Technique RT-PCR		Total
	Positive Samples	Negative Samples	
NEOKIT positives	126	0	126
NEOKIT negatives	13	53	66
TOTAL	139	53	192
True Positives (TP) = 126 (NEOKIT positives that were also positive in RT-qPCR). False Positives (FP) = 0 (NEOKIT positives that were negative in RT-qPCR). True Negatives (TN) = 53 (NEOKIT negatives that were also negative in RT-qPCR). False Negatives (FN) = 13 (NEOKIT negatives that were positive in RT-qPCR). PPV = TP/(TP + FP) = 126/(126 + 0) = 1.000 = 100%NPV = TN/(TN + FN) = 53/(53 + 13) = 0.803 = 80.3%Sensitivity = TP)/TP + FN = 126/(126 + 13) = 0.9067 Specificity = TN/TN + FP = 53/(53 + 0) = 1			

## Data Availability

The data presented in this study are available in the article or Appendix A.

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
