# Peer review of "Colorimetric RT-LAMP Detection of Multiple SARS-CoV-2 Variants and Lineages of Concern Direct from Nasopharyngeal Swab Samples without RNA Isolation"

_viruses, 2023, doi:10.3390/v15091910_

Round 1
Reviewer 1 Report
The reviewed manuscript is dedicated to the design and validation of colorimetric RT-LAMP assay detecting SARS-CoV-2. The presented results are interesting for scientists, specializing on the field of molecular diagnostics. However, a number of issues needs to be addressed before publication.
Major issues:
1. The manuscript is dedicated to design and validation of a new visual LAMP for SARS-CoV-2 detection. However, authors focused the Introduction on description of SARS-CoV-2 variants. Authors are encouraged to add in the Introduction section information about various methods for visual detection of LAMP results and visual LAMP tests for SARS-CoV-2 detection. There are multiple papers describing such tests, and they deserved to be mentioned. Comparison with these previously published tests is also necessary in the Discussion section.
2. Authors are encouraged to stress advantages of their assay over other similar tests and explain why several regions of SARS-CoV-2 genomic RNA were used as LAMP targets.
3. Commonly, in respiratory tests, a control set of primers is applied targeting host DNA or RNA to ensure the presence of genetic material in the probe. Was the same strategy applied in the designed method?
4. Authors are encouraged to provide RT-qPCR Cq values for samples designated as false-negative by RT-LAMP in clinical validation. Plausibly, these samples could contain a low concentration of SARS-CoV-2 RNA.
Minor issues:
1. Careful correction of the language is needed.
2. Page 2, line 47: “to bind six regions” — technically, 8 regions, as each inner primer has two hybridization sites.
3. Page 2, lines 57-63 — the whole paragraph can be moved to the end of the Introduction section.
4. Page 3, line 105: “from Macrogen Inc, Koreaand”
5. Page 4, line 110: “sequence from gene E.with”
6. Sample Inactivation and treatment for direct assay — what transport media was used for swabs storage?
7. Page 4, line 119: “0.1 mM and lysis (buffer B Tris(2-carboxyethyl)”
8. RT-LAMP assay — authors are encouraged to indicate primer concentrations in μM instead of pmol.
9. Page 4, line 130: “the template(30 µl reaction”
10. Page 4, line 132: “color change, from violet”
11. Page 5, line 160: “sets,192 samples”
12. Page 5, line 182: “annealing, Reactivity”
13. Evaluation of primer annealing with SARS-CoV-2 variants and mutations — authors are encouraged to provide a figure with alignment of LAMP primers and different viral variants.
14. Page 7, line 236: “to 98 C.A test”
15. Page 8, line 256: “for |COVID-19”
Please, find comments about the language in the section above.
Author Response
Dear Reviewer 1,
I am sending the new version of the manuscript. This has been thoroughly revised. I greatly appreciate your collaboration.
Best regards,
Adrian Vojnov

Reviewer 2 Report
This manuscript by Werbajh et al. describes the construction of RT-LAMP for detection of SARS-CoV-2 and the evaluation of its sensitivity and specificity. This RT-LAMP system can detect SARS-CoV-2 at least 50 copies/reaction and a variety of variant strains with high specificity. These results are useful findings leading to an accurate and rapid diagnosis of COVID-19.
There are comment as follows.
Major comment:
The sentence structure of the Introduction should be reconsidered. What is written in the conclusion should not be written. Rewrite lines 57-63 and place them at the end of the Introduction. Lines 64-88 should be written in a straightforward manner, deleting unnecessary information.
In an experiment using nasopharyngeal swab, 13 samples that were positive by RT-PCR were negative by RT-LAMP. Can the authors discuss these 13 specimens based on the Ct value or copy number of RT-PCR?
Minor Comment:
Line 15: COVID-19 (Coronavirus disease-19) -> Coronavirus disease-19 (COVID-19)
Line 36: SARS-CoV-2 (Severe Acute Respiratory Syndrome-related Coronavirus - 2) -> Severe Acute Respiratory Syndrome-related Coronavirus - 2 (SARS-CoV-2)
-> There are other places where abbreviations are inappropriate. Reread the manuscript carefully.
Line 145: Are these pathogen samples RNA? If so, the extraction method should be described.
Figure 3: Please clarify whether the two tubes were done with duplicate or two independent test. It should also be described that the three primer sets were mixed.
Table 2: Sensitivity and specificity should be included in Table. 
Round 2
Reviewer 1 Report
Many thanks to authors for their efforts in editing of the manuscript and replies to all comments. However, if possible, language polishing would be highly appreciated, as some synonyms seems do not fit in well in the manuscript.
Language polishing would be highly appreciated, as some synonyms seems do not fit in well in the manuscript.
Author Response

(The authors gave the same response as above.)

Reviewer 2 Report
acceptable
Author Response
Dear Reviewer 2,
I am sending the new version of the manuscript. This has been thoroughly revised. I greatly appreciate your collaboration.
Best regards,
Adrian Vojnov